# Lagrangian surface drifter observations in the North Sea:
# An overview on high resolution tidal dynamics and surface currents

Lisa Deyle[1], Thomas H. Badewien[1], Oliver Wurl[1], Jens Meyerjürgens[1]

[1]Center for Marine Sensors, Institute for Chemistry and Biology of the Marine Environment, Carl von Ossietzky Universität Oldenburg, Oldenburg, 26129, Germany

*Correspondence to*: Lisa Deyle (lisa.deyle@uni-oldenburg.de)

**Abstract.** A dataset of 85 Lagrangian surface drifter trajectories covering the central North Sea area and the Skagerrak from 2017 – 2021 of 17 deployments is presented. The data have been quality controlled, uniformly structured and assimilated in a standard NetCDF format (https://doi.org/10.1594/PANGAEA.963166, Meyerjürgens et al., 2023b). Using appropriate methods presented in detail here, surface currents were calculated from the drifter position data. Based on a drifter deployment in the Skagerrak, it is demonstrated that the Lagrangian measurements can be converted into an Eulerian representation by calculating mean current velocities. Tidal energy spectra were analyzed separately for the southern and northern areas of the North Sea, and tidal ellipses were calculated to determine the tidal impact on surface currents. Significant differences between the shallow shelf and the deeper areas of the North Sea are evident. While the shallow nearshore areas are dominated by tidal currents, deeper areas such as the Skagerrak register a high mean residual circulation driven by high-density gradients. Measurements using Eulerian approaches and remote sensing methods are restricted in temporal and spatial coverage, in particular, to capture fine-scale dynamics. For this reason, Lagrangian measurements, to a large extent, provide new insights into the complex submesoscale dynamics of the North Sea. Exemplarily, the Skagerrak region is used to demonstrate that high resolution drifter observations capture both mesoscale and small-scale current patterns.

This unique dataset covering the entire south-eastern North Sea and the Skagerrak offers further analysis possibilities and can be used for the investigation of various hydrodynamic and environmental issues, e.g., the analysis of submesoscale current dynamics at ocean fronts, the determination of the kinetic eddy energy and the propagation of pollutants in the North Sea.

## 1 Introduction

The North Sea is one of the most intensely used coastal areas worldwide, connecting the major ports of Europe with the world oceans. The region is of particular interest to various stakeholders due to the economic and industrial value of its numerous riparian states. Industrial coastal activities, busy shipping lanes and growing offshore activities in the context of power generation with offshore wind farms increase the anthropogenic pressure on the marine ecosystem.

The North Sea is a semi-enclosed shelf sea opening in the north into the Norwegian Sea and in the western part into the English

Channel. It is highly influenced by freshwater runoffs of the rivers Forth, Humber, Thames, Seine, Meuse, Scheldt, Rhine/Waal, Ems, Weser, Elbe and Glomma (Quante and Colijn, 2016). Due to its shallow water depth in the majority of the area, it is highly impacted by strong tidal dynamics (Otto et al., 1990). Tides influence turbulence intensities, and the energy cascades in shallow areas can create strong vertical mixing, enhancing the formation of tidal mixing fronts in the entire water column (Otto et al., 1990; Ricker et al., 2021). Furthermore, it was shown that tidal motions have a scale-dependent influence

on the relative diffusivities in tidal-controlled shelf environments (Meyerjürgens et al., 2020). However, the particle motions are controlled by both the tides and the residual currents, the latter are mainly caused by density gradients and wind.

Numerical models depend on in situ measurements to be developed further and for calibration purposes (Ricker et al., 2021; Ricker and Stanev, 2020). From 1970 to 1990, many investigations on tidal dynamics and surface currents, and thus the transport in the North Sea, have been reported in the literature (Becker et al., 1999; Otto et al., 1990). A dense network of

monitoring stations has been established in the North Sea region in the last decades, which makes it one of the best-observed coastal areas worldwide (Baschek et al., 2017). Nevertheless, further investigations are valuable because today's technical progress offers a higher quantity and quality of data and, in terms of spatial distribution, there still exist significant data gaps in the North Sea (Sündermann and Pohlmann, 2011). Previous studies of surface currents and tidal dynamics in the North Sea have consisted of mooring arrays with current meters that have taken long-term measurements using Eulerian measurement

techniques (Davies and Furnes, 1980; Maas and van Haren, 1987). Other applications cover Acoustic Doppler Velocity Profilers (ADCP) mounted on ferries to gather current data on ferry routes (Vindenes et al., 2018). Similarly, near-shore high-frequency radars or satellite altimeters have been used to study surface currents (Baschek et al., 2017). One major drawback of all these measurement techniques is their focus on fixed points of regions covering distinct timescales, or in the case of satellite altimeters, measurements are continuous in space and time, but with coarse resolution.

Lagrangian measurement techniques, which have been rarely used in the North Sea (Meyerjürgens et al., 2019), track the horizontal propagation of ocean currents by following their pathways over long time and spatial scales. With high sampling intervals and long-term measurements, Lagrangian techniques can capture small spatially and temporally rapid movements, as well as large-scale temporal variations (Lilly and Pérez-Brunius, 2021a; McWilliams, 2016). Lagrangian measurements can cover mesoscale ocean circulations, small-scale and submesoscale fluid dynamics to improve the understanding of complex

surface current and energy dynamics (Lumpkin et al., 2017; Özgökmen et al., 2012). Differential kinematic properties and dispersion characteristics of the submesoscale current field can be determined, allowing an understanding of the structure and

dynamics of the ocean surface, for example, at density fronts (Essink et al., 2020; Tarry et al., 2022). Furthermore, several studies analyzed the propagation of particles and Lagrangian surface drifters at the sea surface to understand the ecological implications of plastic litter (Van Sebille et al., 2020; Meyerjürgens et al., 2023a) and oil spills (Liu and Weisberg, 2011) in open ocean and coastal areas. For this reason, a satellite-tracked high-resolution surface drifter, which is adjustable with various wind slip characteristics, was developed that follows surface currents at 0.5 m depth and can be used in tidally influenced coastal and estuarine areas (Meyerjürgens et al., 2019) as well as in open ocean environments (Martín et al., 2023).

This study presents drifter data collected from 2017 to 2021 in the North Sea. Previously, only individual datasets have been analyzed for individual scientific objectives. This paper provides a comprehensive Lagrangian drifter dataset that presents an overview of the coverage of all collected data, contributing to a better understanding of the circulation and tidal dynamics in the North Sea. It is presented for the first time in the North Sea due to a significant lack of studies using drifters in the past. The particular focus will be on presenting the dataset and describing the processing methods in detail.

## 2 Study Area

The North Sea is a semi-enclosed shelf sea surrounded by Great Britain and central northern continental Europe. At the northern edge, it is connected to the Norwegian Sea and the North Atlantic by a broad opening, whereas in the southwest, it is narrowly linked by the English Channel to the North Atlantic (Fig. 1). In addition, to the high saline Atlantic water, the North Sea gains a substantial inflow of the low saline Baltic water and freshwater inflows from river runoffs (Otto et al., 1990).

The average depth of the North Sea is around 80 m, becoming shallower from north to south toward the coast (Otto et al., 1990; Vindenes et al., 2018). The Norwegian Trench and the Skagerrak, as an extension, are the deepest regions with a sill depth of approximately 270 m and a maximum depth of 700 m (Vindenes et al., 2018).

The tides in the North Sea are initiated by tidal waves from the North Atlantic. This emerges from the Norwegian Sea, moves along the coast of Great Britain and propagates around three amphidromous points in the North Sea (Vindenes et al., 2018). These are located at the southwestern tip of Norway, at the eastern tip of Dogger Bank, and near the entrance to the Southern Bight (Otto et al., 1990; Vindenes et al., 2018). The dominant tidal component is the tidal motion of the semidiurnal $M_2$ tide and provides a good initial approximation of the tidal motion. Tidal currents can reach maximum velocities above 1 m s$^{-1}$ for spring tides in the shallow southern part of the North Sea and along the coast of Great Britain (Dietrich, 1950; Valle-Levinson et al., 2018). In the deeper northern part, maximum velocities of 0.2 – 0.4 m s$^{-1}$ are reached (Otto et al., 1990), and lowest in the Skagerrak with a magnitude of 0.01 m s$^{-1}$ (Danielssen, 1997; Rodhe, 1987).

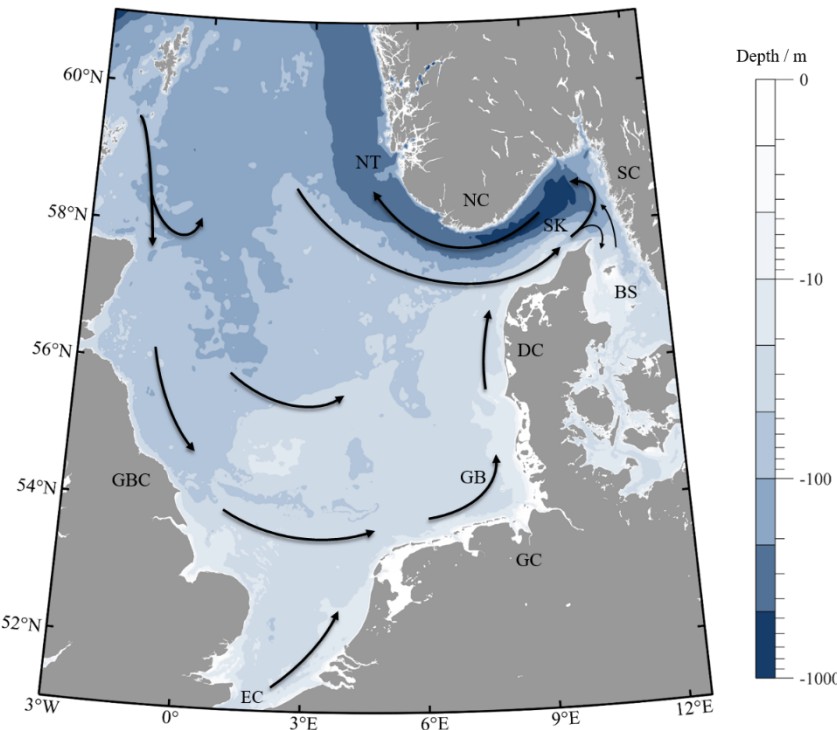

**Figure 1: A map of the North Sea and its surroundings, showing the major mean ocean circulation pattern in the North Sea. Acronyms: Baltic Sea (BS), Danish Coast (DC), German Bight (GB), Great Britain Coast (GBC), German Coast (GC), Norwegian Coast (NC), Norwegian Trench (NT), Swedish Coast (SC), Skagerrak (SK) (based on Otto et al. (1990)). The bathymetry of the North Sea is shown in blue.**

In addition to the cyclic tidal components such as $M_2$, nonlinear advection terms occur in the shallow coastal regions of the North Sea, which become more effective with decreasing water depth (Stanev et al., 2015). Even harmonic shallow water overtides are generated, such as the $M_4$ tide, which occurs twice as often as the semidiurnal $M_2$ tide, deforming the original sinusoidal character of the tidal currents (Otto et al., 1990; Stanev et al., 2015). Furthermore, odd harmonics (e.g., $M_6$) are generated due to bottom friction (Le Provost, 1991; Stanev et al., 2016). An interaction of the different tidal components causes tidal asymmetry and can be essential for tidal dynamics in shallow water areas. Other residual currents are due to prevailing westerly winds and varying density gradients due to freshwater inputs, mainly from the Rhine, Ems, Weser, and Elbe rivers (Otto et al., 1990; Ricker et al., 2020).

The residual currents lead to a permanent displacement of water masses, forming a cyclonic residual circulation (Otto et al., 1990; Burchard and Badewien, 2015). Thus, the circulation turns eastward at the southern part of the coast of Great Britain, flows further along the coast of Germany, and then flows northward along the Danish coast into the Skagerrak (Fig. 1). It should be noted that strong, persistent winds from the south and east can weaken or even reverse the circulation pattern (Howarth, 2001). In the shallow area of the North Sea, the surface currents are dominated by the tidal components (Otto et al., 1990). In contrast, in deeper areas, such as the Skagerrak, the surface currents are driven by strong density differences resulting

from strong inflows from the North Atlantic, North Sea, Baltic Sea, and river runoffs, so the residual circulations are more extensive in this area (Howarth, 2001).

## 3 Material and Methods

The dataset presented here was obtained using GPS (Global Positioning System) data from satellite-tracked surface drifters, whose state-of-the-art design is compact, cost-effective, and light-weight (Fig. 2). The drifters are designed to follow the upper 0.5 m surface currents by four cruciform drag-producing vanes to minimize the direct wind slip effect. The drag area ratio, which is a measure for the direct wind slip, was calculated by Meyerjürgens et al. (2019) accounting for 0.27 % resulting in wind-induced velocities ranging from $0.0027 - 0.027$ m s$^{-1}$ by wind speed amplitudes of $1 - 10$ m s$^{-1}$.

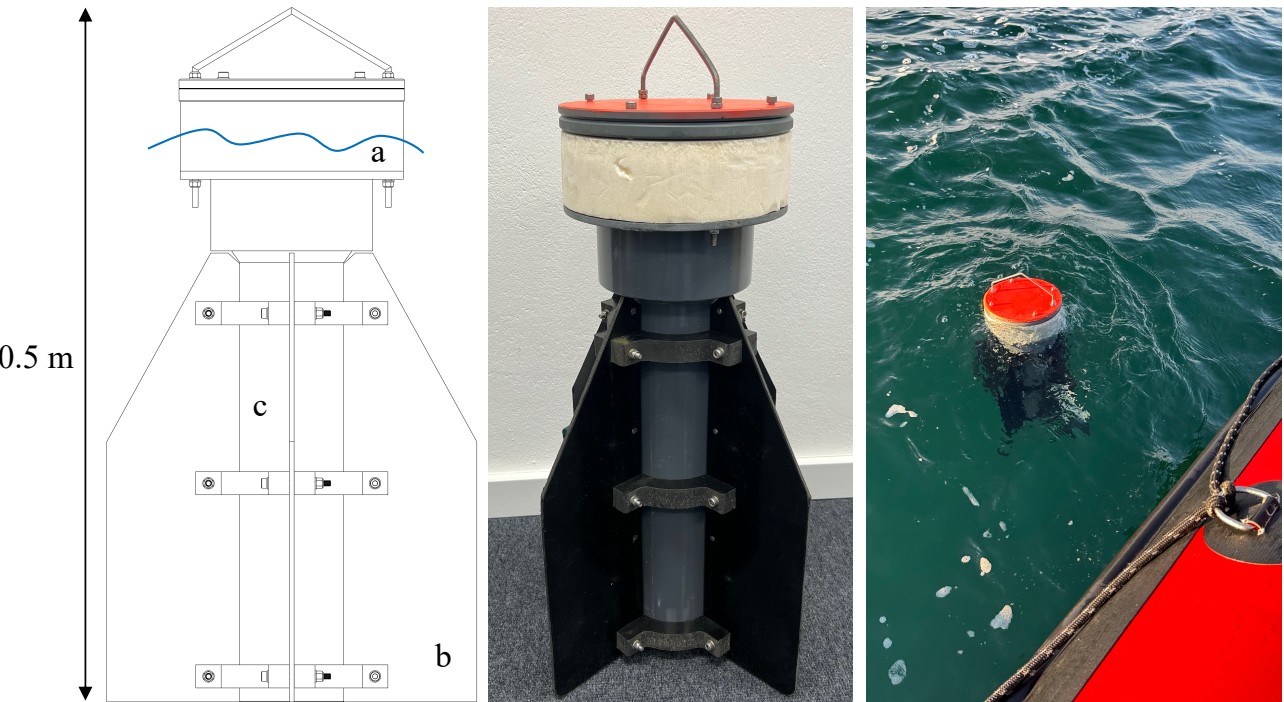

**Figure 2: Left: a drawing of the Lagrangian high resolution surface drifter: (a) buoyancy of the housing containing the GPS unit, (b) drag-reducing wings, (c) lower part of the housing with battery pack and ballast. It has a height of 0.5 m and floats at the height of the sketched blue wave on the sea surface; center: a photo of an assembled drifter; right: the drifter during a deployment.**

The upper part of the housing contains a cost-effective, commercial off-the-shelf GPS receiver (Spot Trace®), which offers a good global coverage of 96 % (Meyerjürgens et al., 2019). Numerous surface drifter experiments with Spot Trace® GPS

receivers in the Gulf of Mexico have demonstrated the successful use of the positioning and telemetry module (Novelli et al., 2017). With selectable intervals of 5, 10, 30, or 60 minutes, the Spot Trace® transmits the position of the drifter in near-real time via the Globalstar® satellite network (Meyerjürgens et al., 2019).

The upright position of the drifter is ensured by a Styrofoam ring in the upper part of the housing, as well as a battery pack and weights in the lower part of the drifter (Meyerjürgens et al., 2019). The battery pack consists of up to 12 D-cell alkaline
batteries which supply the Spot Trace® with power and extend the operating time of the drifter to about 9 months.

Due to the high sampling rate and the long runtime, both short- and long-term datasets can be analyzed. In addition, in contrast to many other drifters, the compact size and wing design enable the deployment of this drifter in shallow water environments and its resuspension in coastal areas after it has washed ashore. A detailed description of the drifter can be found in Meyerjürgens et al. (2019).

This section describes the data processing and quality control and further presents the methods to analyze the tidal dynamics and the surface current fields. First, the various deployments of the drifters are presented (Sect. 3.1). In Sect. 3.2, the drifter data processing, including quality control, outlier removal, and interpolation schemes are presented. In the next section, the method used to calculate the drifter velocities and the residual currents is explained (Sect. 3.3). Finally, in Sect. 3.4 the analysis of the tidal dynamics, including the calculation of the power spectral density, is given.

**3.1 Drifter Deployments**

The drifters were deployed on different research expeditions from 2017 to 2021 (Tab. 1).

**Table 1: Information for the surface drifter datasets in the North Sea. From left to right: name of experiment; deployment area; original sample interval in minutes; number of drifter trajectories; number of 5-minute data points after interpolation; percentage of how often drifters did not transmit over an hour; date of first data point within a deployment; date of last data point within a**
**deployment; mean duration of a deployment in days; plus or minus the standard deviation of trajectory durations in days; maximum trajectory duration in days. Datasets were sorted by year (based on Lilly and Pérez-Brunius (2021a)).**

| Surface drifter data from the North Sea | | | | | | | | | |
|---|---|---|---|---|---|---|---|---|---|
| Name | Area | Interval / min | Traj. | Points | Fill / % | First Date | Last Date | Duration / d | Max / d |
| Mar17 | German Bight | 10 | 6 | 39188 | 0.304 | 13-Mar-2017 | 22-Apr-2017 | 23±14 | 40 |
| Aug17 | German Bight | 10 | 2 | 7501 | 0.536 | 02-Aug-2017 | 24-Aug-2017 | 13±13 | 22 |
| Oct17 | German Bight/Southern Denmark | 10 | 5 | 17391 | 0.282 | 08-Oct-2017 | 27-Oct-2017 | 12±4 | 15 |
| Feb18 | North Sea | 10 | 6 | 164636 | 0.239 | 24-Feb-2018 | 25-Jun-2018 | 95±19 | 116 |
| Oct18 | German Bight/Denmark/Norway | 10 | 15 | 265168 | 1.373 | 21-Oct-2018 | 22-Jan-2019 | 61±22 | 93 |
| Mar19 | Southern German Bight | 10 | 7 | 8896 | 0.483 | 22-Mar-2019 | 27-Mar-2019 | 4±0 | 5 |
| Jul19 | German Bight/Denmark | 10 | 11 | 94993 | 0.638 | 12-Jul-2019 | 12-Aug-2019 | 30±1 | 31 |
| Jul21 | German Bight | 5 | 6 | 48636 | 0.157 | 22-Jul-2021 | 25-Aug-2021 | 28±4 | 35 |
| Sep21 | (Eastern) German Bight | 5 | 10 | 45066 | 0.122 | 12-Sep-2021 | 05-Oct-2021 | 16±6 | 23 |
| Oct21 | Skagerrak | 5 | 17 | 117635 | 0.214 | 07-Oct-2021 | 24-Nov-2021 | 24±14 | 48 |
| ALL | North Sea | 5 | 85 | 809110 | 0.435 | 13-Mar-2017 | 24-Nov-2021 | 31±10 | 116 |

They are equipped with a GPS tracker that transmits the coordinates of the drifter position and time by satellite. Depending on the measurement strategy a transmission interval of 5 or 10 minutes was set. As a result, 85 high-resolution trajectories with 85 surface drifters could be measured during the entire period, which are shown in Fig. 3.

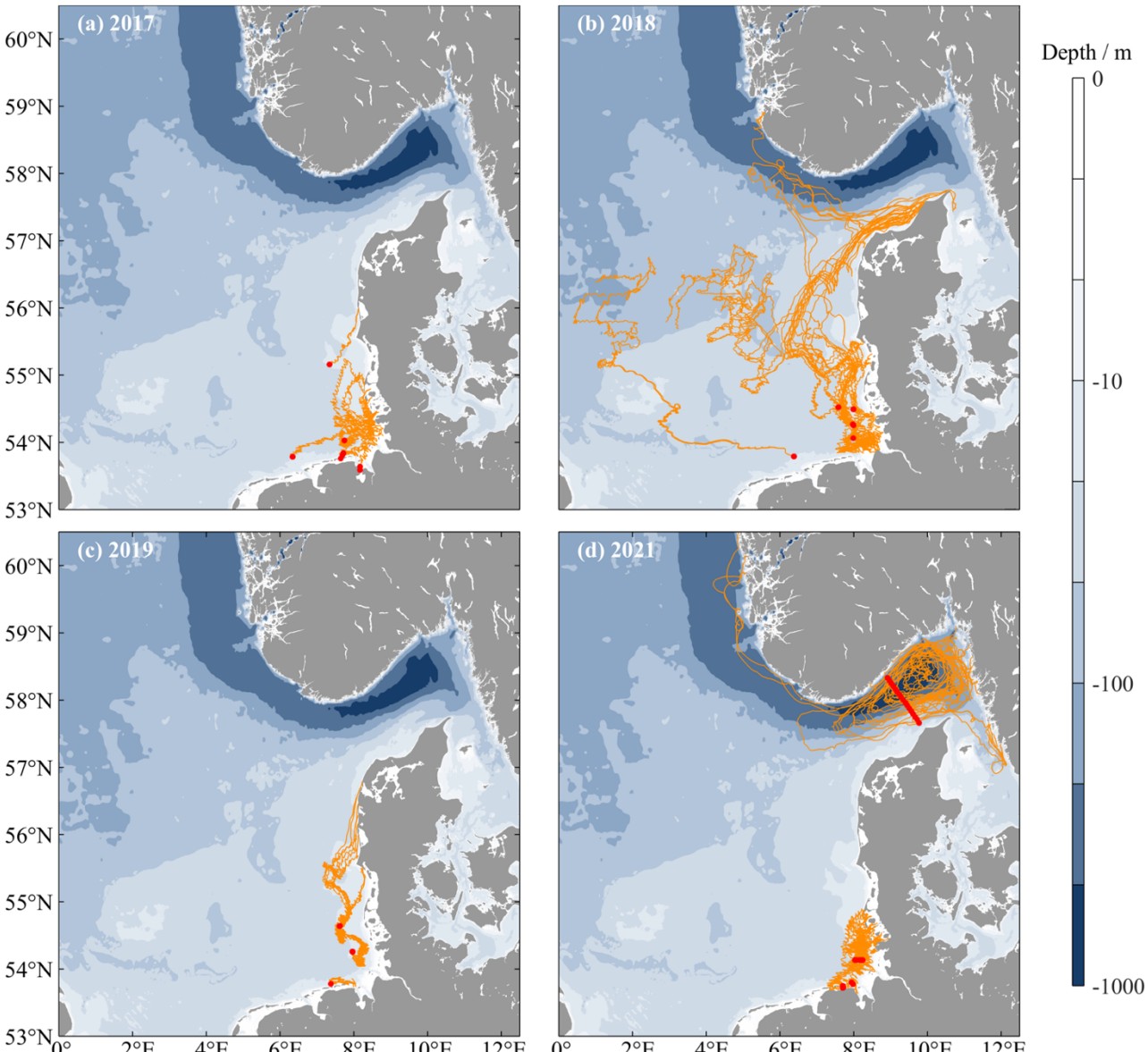

**Figure 3: Processed and interpolated surface drifter data in the North Sea from 2017 (a), 2018 (b), 2019 (c) and 2021 (d). The starting points of the trajectories are indicated by red dots.**

In 2017, the drifters were deployed in clusters in different areas of the German Bight (Fig. 3a). Similarly, in 2018, all drifters were deployed in the German Bight, but the measurements captured a larger area because the drifters covered a period of at least 39 days (Fig. 3b). Many drifters were transported northward into the Skagerrak with the cyclonic residual circulation trend of the North Sea, but due to a strong sustained easterly wind in early 2018, some drifters moved westward (Ricker et al., 2020; Stanev et al., 2019). A short drifter deployment in the southern German Bight was realized in spring 2019, as well as a longer measurement of about 30 days along the German-Danish coast in summer (Fig. 3c). After a break in measurement in

2020, three larger measurement campaigns followed in 2021, two in the German Bight, the third for the first time in the Skagerrak, where 17 drifters were deployed along a transect (Fig. 3d).

**3.2 Data Processing**

In order to use the data, several processing steps had to be taken. First, the raw data were cut so that beachings were removed. In the next step, outliers were eliminated from the latitude and longitude data with a hampel median filter. The filter calculates the median and standard deviation for each sample value for a window of 18 surrounded samples (9 per side). All drifter positions that deviate from the median by more than three standard deviations are replaced by the median in the time series
dataset. In the present study, 0.041 % of the total dataset deviates from the median by more than three standard deviations, indicating that the dataset has only a small number of outliers that had to be replaced. As drifter positions were measured at irregular time intervals, in the final step, all data was interpolated using a piecewise cubic interpolation method (Fritsch and Carlson, 1980), also known as the pchip method. For an overview, the individual interpolated data points, with the 5-minute intervals, were summed to show the total number of data collected by the drifter after the interpolation (Tab. 1). The percent
of filled data describes how often data gaps of more than one hour occurred. These data gaps are caused by data transmission errors, which can occur especially in bad weather conditions. During strong winds and high waves, the drifters are washed over more frequently, which can restrict the communication between the satellites and the GPS trackers of the drifters. The interpolated data is considered 'filled' if no data has been supplied in the raw dataset for more than one hour. This affects, as seen in Tab. 1, an average of only 0.435 % of the collected data. This proves their reliability and the high quantity and high
resolution of data sent by the drifters.

Finally, all data were sorted by assimilating into a standard NetCDF format. The dataset contains drifter position data (*lat*, *lon*, *datenum*, *time*), drifter-derived current velocities (*u*, *v*) and two vectors (*beached*, *filled*) describing when a drifter was beached and when the dataset was filled because no GPS position was sent for more than one hour (Meyerjürgens et al., 2023b). All data are associated with a drifter ID (*drifter_id*) and a deployment ID (*deploy_id*) to identify which data belongs to a drifter
and when multiple drifters have been deployed in a single deployment.

**3.3 Velocity Calculation**

The current velocities can be calculated with a forward difference scheme (Eq. 1, 2):

$$u_d = \frac{x_d(t+\delta t) - x_d(t)}{\delta t} \tag{1}$$

and

$$v_d = \frac{y_d(t+\delta t) - y_d(t)}{\delta t} \tag{2}$$

where $x_d(t)$ and $y_d(t)$ are the zonal and meridional coordinates, $\delta t$ is the time interval and $u_d$ and $v_d$ are the zonal and meridional velocity components.

The first step involves the extraction of the position data for a selected region. This can be done by setting limits for the latitude and longitude values or by selecting the desired expeditions or deployments from the NetCDF structure. In the next step, the longitude and latitude coordinates in degrees must be converted to zonal and meridional coordinates $(x_d(t), y_t(t))$. Afterwards, the forward difference method is used to calculate the zonal $u_d$ and meridional $v_d$ velocity components from the zonal and meridional coordinates determined at time $t$ (Eq. 1, 2). The velocity $cv = u_d + \sqrt{-1} \cdot v_d$ in m s$^{-1}$ is determined from the zonal and meridional velocity components $u_d$ and $v_d$.

It is important to note that the velocity components $u_d$ and $v_d$ include both, the tidal and the residual currents. To represent the mean residual circulations, the tidal component must be extracted. Therefore, a moving average filter is used over a 24.83-hour period, which low-pass filters the velocity time series. With this time period the dominant tidal constituents for the North Sea, such as $M_2$ and $S_2$, are removed. The moving average filter removes these tidal constituents from the zonal and meridional current velocities $u_d$ and $v_d$.

### 3.4 Tidal Analysis

### 3.4.1 Power Spectral Density

The raw velocity time series signal of $u_d$ and $v_d$ are used to observe the tidal effects. Due to their periodic pattern, the tidal currents can be decomposed into their harmonic fundamental components with their representative frequency. If a time series is of appropriate length, the tidal constituents at any location can be determined. The main tidal constituents of the North Sea that have been considered in more detail in other studies are listed in Tab. 2 (Meyerjürgens et al., 2019; Otto et al., 1990; Vindenes et al., 2018). For the representation of tidal currents in the frequency domain, the power spectral density (PSD) is suitable. The power spectral density is used when non-periodic components are present in the signal in addition to periodic components. These signal types are called random signals. This is useful for the signals in this study, since non-linear influences, such as different wind strengths, can also affect the signal. As in a standard Fourier transform, the signal is decomposed into individual frequencies, but at the same time, the power spectral density describes the proportion of different frequencies over the total power and indicates the energy distribution of a signal (Dempster, 2001). The power spectral density is normalized to a single hertz bandwidth, giving a uniform value regardless of the bandwidth.

The first step is to select the region where the tidal currents are to be analyzed. In this study, the latitude ranges < 55° N, 55° N - 57° N and > 57° N were chosen. Otto et al. (1990) divided the North Sea in their study into a southern, a central and a northern section, in which different dynamics are to be expected. The velocity time series are assigned to these regions. Trajectories that are located in multiple regions are divided into individual time series data and assigned to the appropriate region. In addition, a minimum signal length should be established. The velocity time series should be of sufficient length to filter the tidal constituents as accurately as possible. Studies that analyzed tidal components with drifter data used minimum times series between 4 and 40 days (Elipot et al., 2010; Poulin et al., 2013; Callies et al., 2019; Meyerjürgens et al., 2020). It should be noted that the studies with longer time series have a lower sampling rate (>1 hour), than the drifter in this study.

With a higher sampling rate, a shorter time series is required as the tidal components can be more clearly differentiated from the remaining current components. In this study, it is important, that clear peaks are identifiable for the determination of the power spectral density in order to obtain statistically significant results. Similarly, it should be short enough that a large number of time series can be included in an analysis to provide a suitable power spectral density. Since a power spectral density is always decomposed into a respective power of two of the signal length, in this analysis a time series with a minimum of 4096

intervals ($2^{12}$ = 4096) was chosen, which corresponds to a period of about 14.2 days. For each velocity time series of the corresponding length, the power spectral density is determined. In the last step, the individual power spectral densities from the same region are averaged so that a mean power spectral density can be presented for each region (< 55° N, 55° N - 57° N and > 57° N).

**Table 2: Main tidal constituents of the tides in the North Sea. The period is given in hours, frequency in cycles per day.**

| Description | Symbol | Period / h | Frequency / $d^{-1}$ |
|---|---|---|---|
| Principal lunar semidiurnal constituent | $M_2$ | 12.4206 | 1.9323 |
| Principal solar semidiurnal constituent | $S_2$ | 12.0000 | 2.0000 |
| Shallow water overtides of principal lunar constituent | $M_4$ | 6.2103 | 3.8645 |
| Shallow water overtides of principal lunar constituent | $M_6$ | 4.1402 | 5.7968 |
| Shallow water overtides of principal lunar constituent | $M_8$ | 3.1052 | 7.7291 |
| Luni-solar diurnal constituent | $K_1$ | 23.9345 | 1.0027 |
| Lunar diurnal constituent | $O_1$ | 25.8193 | 0.9295 |

**3.4.2 Tidal Ellipses**

Similar to the calculation of the power spectral density, the unfiltered time series of velocities *cv* are taken to calculate and display tidal ellipses. They are used to analyze the individual tidal constituents using the tidal harmonic analysis toolbox *t_tide* by Pawlowicz et al. (2002). It should be noted that only sufficiently long time series are taken. In this study, a length of at least 3000 intervals (10.4 days) was specified, since a Fourier transform above this length yields plausible results with distinct peaks.

Studies by Elipot et al. (2010), Poulin et al. (2013), Callies et al. (2019) and Meyerjürgens et al. (2020), which analyzed time series with lengths between 4 and 40 days, can be used for comparison. The tidal extraction results can be plotted using the tidal ellipse parameters of a tidal constituent provided by *t_tide*. The tidal ellipse is characterized by the major axis constituent *fmaj*, the minor axis constituent *fmin*, and the ellipse orientation *finc*, which describes the inclination angle (Fig. 4). The major axis represents the maximum and the minor axis the minimum tidal current velocity of the respective tidal component. The

ellipse orientation describes the angle between the eastward direction and the major axis counterclockwise. In the last step, all tidal ellipse parameters of the individual time series are averaged so that an averaged ellipse is determined for an area. This study focuses on the analysis of the lunar semidiurnal $M_2$ tide, as this tide is the most dominant tidal component of the North Sea. Other tidal components can be computed using the same procedure.

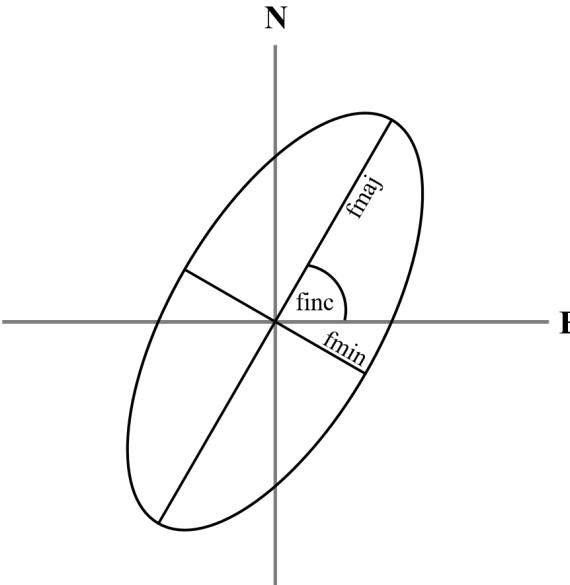

**Figure 4: Illustration of a tidal ellipse.** *fmaj* **is the major axis,** *fmin* **the minor axis, they describe the maximum and minimum velocity**
**of the tidal currents.** *finc* **is the inclination of the ellipse and describes the angle between the east direction and the major axis**
**counterclockwise (based on Vindenes et al. (2018)).**

## 4 Results

The trajectories in Fig. 3 reflect the circulation pattern in the North Sea. Southern drifters, for example from spring 2019
(Fig. 3c), are transported eastward along the northern German islands. The drifters in the eastern German Bight are transported
along the Danish coast into the Skagerrak. In this area, the drifters spread with the inflow from the Jutland Current, or along
the northern slope in the opposite direction with the Norwegian Coastal Current. Individual drifters from 2018 were transported
contrary to the trend of the mean residual currents to the west and not into the Skagerrak. This is due to extreme events in the
North Sea with strong sustained easterly wind during this period (Ricker et al., 2020; Stanev et al., 2019) (Fig. 3b).

For a statistical overview, the number of drifter observations per cell was calculated in Fig. 5. For the data, a cell resolution of
0.125° in latitude and longitude direction was chosen. In the German Bight, it is apparent that there are up to more than $10^4$
drifter measurements per cell. To the west and north of the German Bight, the values are significantly lower, but still mostly
comprise $10^2 - 10^3$ values per cell.

In addition to the drifter position data, the dataset contains the current velocities. These are highly variable depending on the
region and season. Using the method from Sect. 3.3, the residual surface velocities indicate that the Skagerrak area is
characterized by strong currents caused by winds and the inflow from the Jutland Current and the outflow from the Baltic Sea
and the Norwegian Coastal Current. Almost the entire area of the Skagerrak was covered within a deployment in October and
November 2021. Due to the seasonal deployment, the velocity data can be converted into an Eulerian representation for an
overview. The individual velocities are assigned to the corresponding cell and an average value is calculated for each cell;

cells with data gaps are not displayed (Fig. 6a). The arrows indicate the directional components of the mean residual velocities

of $u$ and $v$, and the length describes the magnitude of the velocity. A cell resolution of 5 km in latitude and longitude direction was chosen. The observations per cell with the drifter trajectories in the background are shown in Fig. 6b.

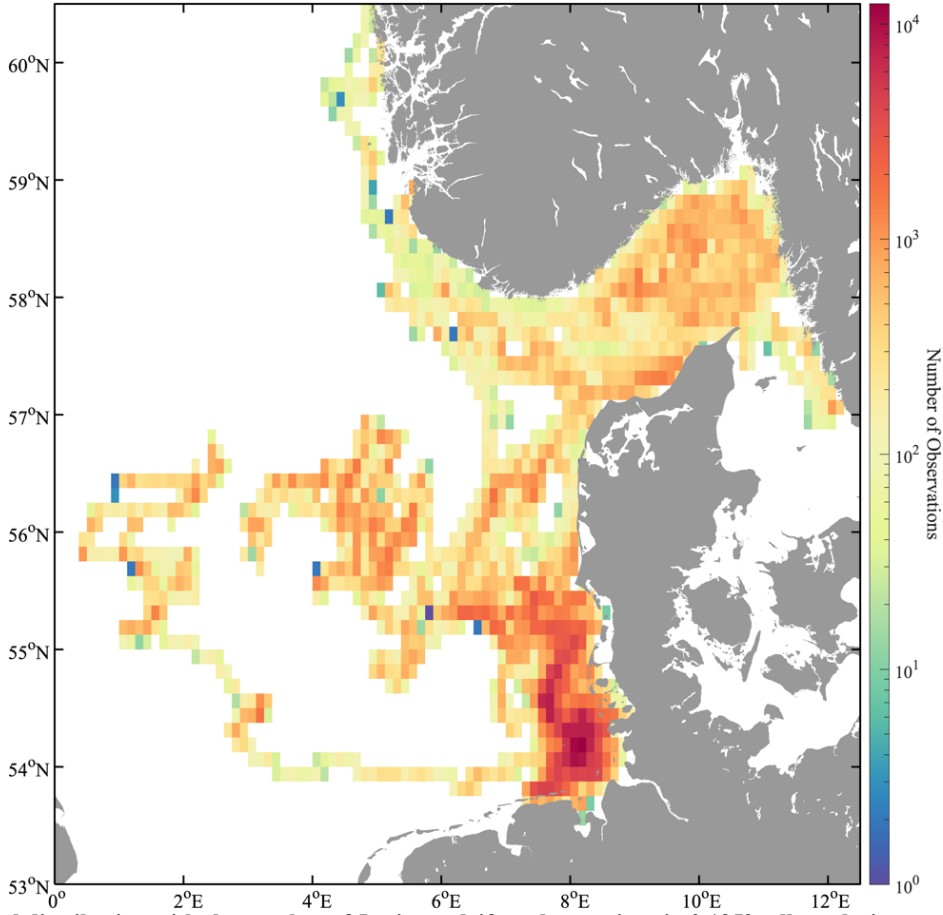

**Figure 5: The spatial distribution with the number of 5-minute drifter observations in 0.125° cell resolution on a logarithmic scale.**

The Jutland Current with North Sea water (orange-red) flowing eastwards and the Norwegian Coastal Current (yellow) flowing in the opposite direction are strongly visible with their velocities of $0.5 - 0.8$ m s$^{-1}$ (Fig. 6a). It should be noted that maximum

values of up to 1.0 m s$^{-1}$ were measured in a small number of cells. Changes in direction at the Norwegian-Swedish coast and the outflow into the Baltic Sea, are clearly visible.

The number of observations demonstrates that even with a cell resolution of 5 km, the majority of the measured drifter data provide more than 50 values per cell (Fig. 6b). This provides a high resolution seasonal snapshot of the mean currents in the Skagerrak. Higher resolutions are required in order to be able to resolve finer current processes. Between the Jutland Current

and the outflow of the Norwegian Coastal Current, the small processes of recirculation can be resolved with a high cell resolution.

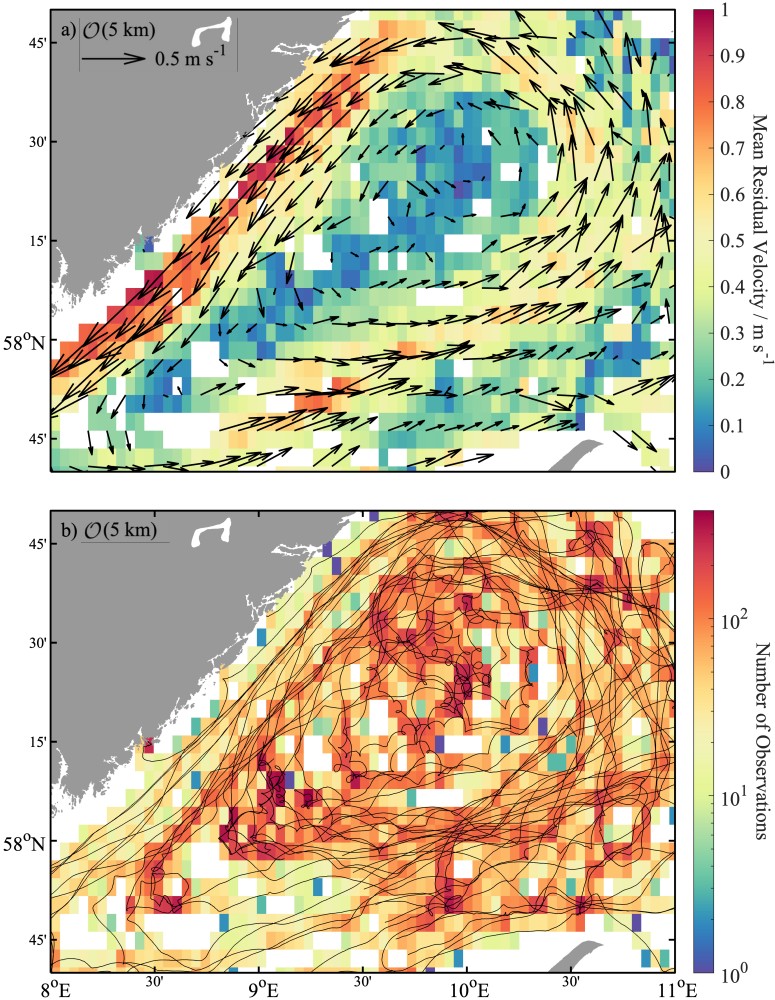

**Figure 6: Mean velocities of the residual circulations in the Skagerrak with 5 km cell resolution (a). The arrows indicate the direction of the mean velocities, the magnitude describes the strength of the velocity in m s$^{-1}$; The spatial distribution with the number of 5-minute drifter observations in 5 km cell resolution on a logarithmic scale (b). The drifter trajectories are displayed in the background.**

Current differences in the North Sea between the shallow shelf and the deeper regions can be seen, for example, in the tidal ellipses of the semidiurnal M$_2$ tide (Fig. 7). For each blue ellipse, a latitude range of 2° was considered between 4° E - 9° E and for each red ellipse, a longitude range of 3° between 57° N - 60° N. The tidal dominance in the shallow water shelf is evident. It can be concluded that the shallower the North Sea and the greater the distance to the three amphidromous points, the higher the mean tidal currents become. Maximum mean current velocities are observed of up to 0.49 m s$^{-1}$ in the southern German Bight and 0.12 m s$^{-1}$ around the Danish coast. Furthermore, the elliptical orientation of the tide decreases from 145.4° to 123.6°, which means that the north-south motion slightly increases. In the deep North Sea, the tidal wave has little effect on the circulation. In the Norwegian Trench only velocities of 0.013 – 0.065 m s$^{-1}$ are reached, in the Skagerrak velocities of 0.007 – 0.041 m s$^{-1}$.

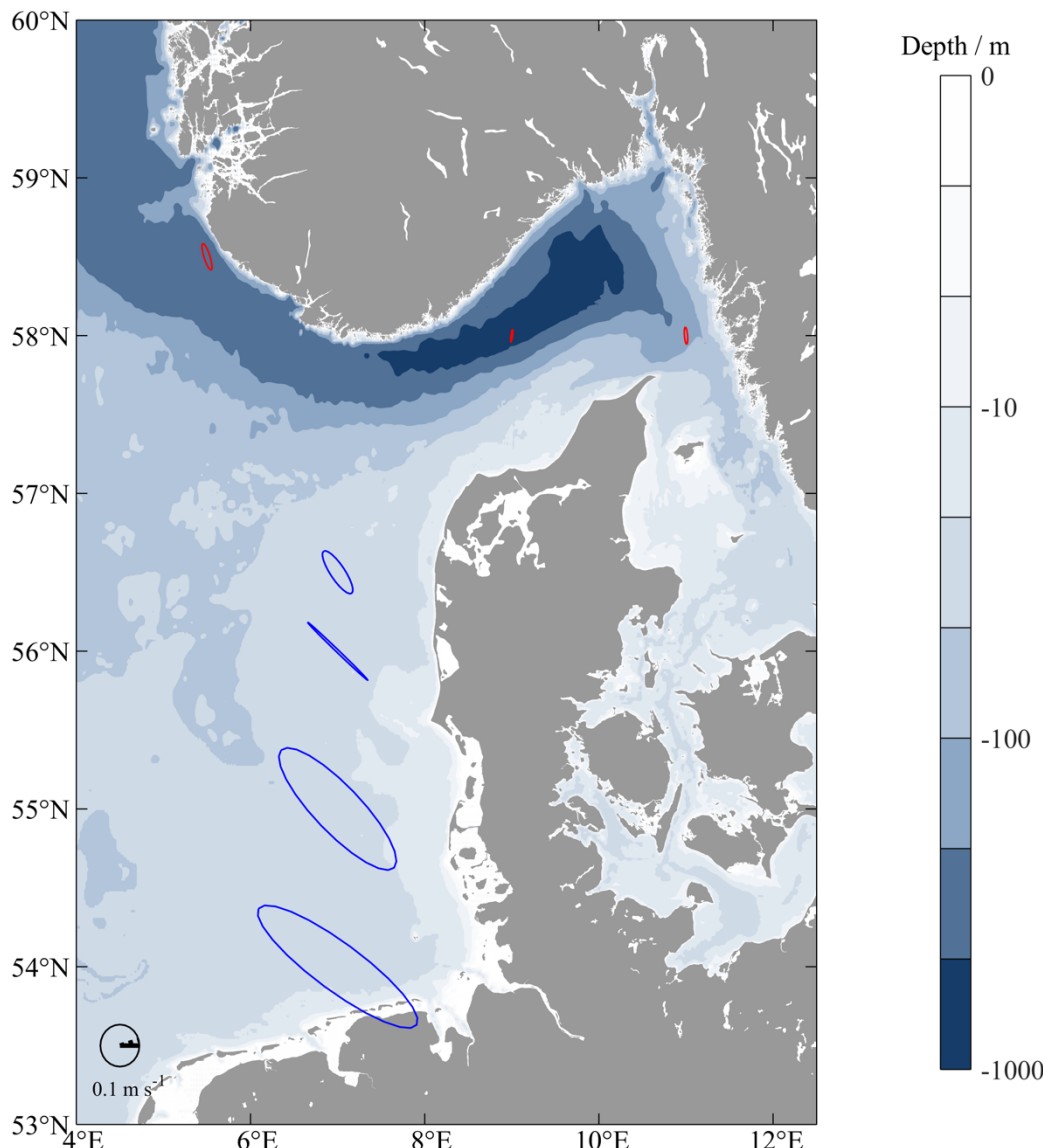

**Figure 7: Calculated mean tidal ellipses for the North Sea. The colours of ellipses describe the size of the cells used for the calculation: blue: 2° lat, 5° lon; red: 3° lat, 3° lon.**

In addition, the analysis of the mean power spectral density provides further insights into the tidal analysis (see Sect. 3.4.1). The zonal component shows that in the southern region of the North Sea ($< 55°$ N) the tides are most dominant (Fig. 8a). The peak of the dominant tidal constituent $M_2$ with 1.93 cycles per day is apparent. Furthermore, the shallow water tidal 285 components $M_4$, $M_6$ and $M_8$ are shown with smaller peaks, as well as the solar semidiurnal tide constituent $S_2$. In the latitude range of $55°$ N - $57°$ N, the peaks of the zonal $M_2$, $M_4$, and $M_6$ tides are also clearly visible, but the magnitude of $M_6$ is smaller. In the Skagerrak area, a weak elevation can be recognized in the area of the $S_2$ and $M_2$ tidal components, but no distinct peaks

can be identified. The mean power spectral densities of the meridional velocity time series show similar characteristics, but the peaks are not as dominant as those of the zonal mean power spectral densities and no peak is present in the frequency range of $M_8$ (Fig. 8b). Despite the fact that the overall meridional tidal motion is smaller, it increases towards the north (55° N - 57° N). In the deep area of the Skagerrak, the tidal currents are also not different from the noise for the meridional components.

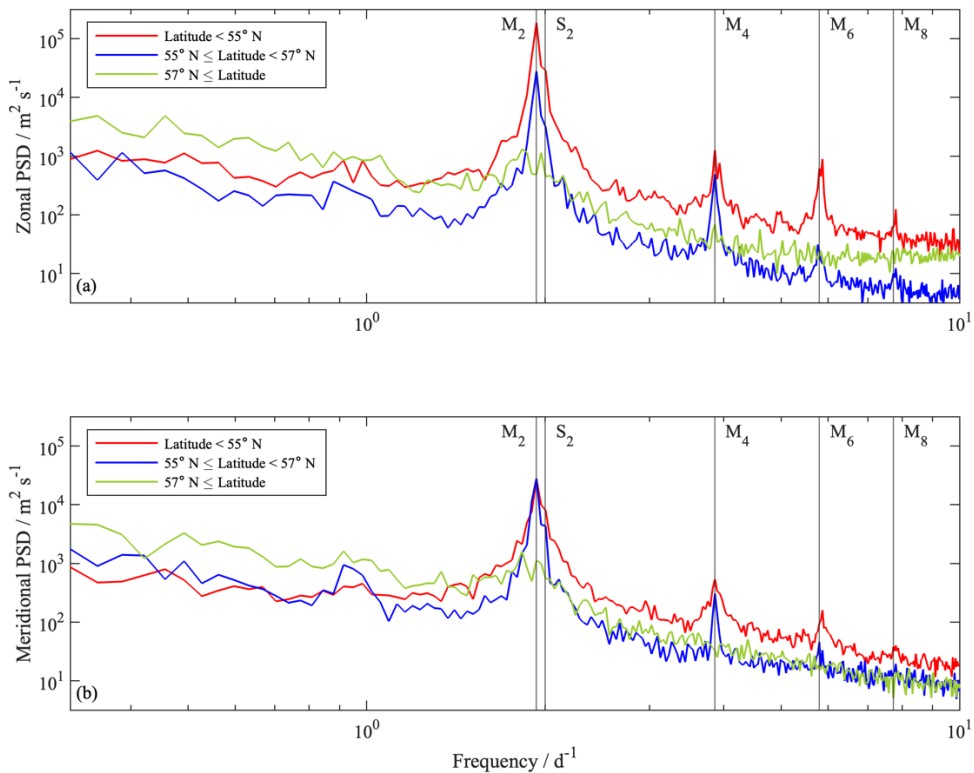

**Figure 8: Mean power spectral density on (a) the zonal *u* and (b) the meridional *v* drifter velocities in the North Sea averaged over all trajectories. The periods of the semidiurnal and shallow water tidal components are indicated by the vertical lines.**

## 5 Discussion

In this study, a comprehensive Lagrangian dataset is presented for the North Sea region for the first time. The dataset is valuable for various kinds of marine science research topics, e.g., marine litter transports and carbon or nutrient exchange processes, and provides important insights into the tidal dynamics and complex surface current patterns in the region. Lagrangian surface drifters offer the ability to cover different areas with a single system because they move with the current. It is a simple and cost-effective system and is completely autonomous after being deployed. In the past, the coverage of the entire North Sea in terms of current patterns has been only provided by satellite and remote sensing methods, but these techniques cannot provide the temporal and spatial resolution to understand the smaller-scaled and highly dynamic processes in the North Sea. Therefore, satellite and radar remote sensing observations are often combined with in situ platforms, resulting in the further development

of model data for the North Sea. Drifter trajectories, for example, can be ideal subjects to be compared with model tracer trajectories in a one-to-one comparison study, or they can be used to derive dispersion properties. Studies by Baschek et al. (2017), Ricker and Stanev (2020), Stanev and Ricker (2020) and Ricker et al. (2021) have used drifter trajectories for comparison studies and model validations. For this reason, Lagrangian surface drifters provide platforms for area-wide in situ measurements and are essential to support and promote further development of model analysis and particle tracking approaches. Lagrangian in situ measurements also benefit from the advancement of models. For example, the detection of tidal mixing fronts in the North Sea is currently challenging due to the high dynamics. More accurate prediction models would facilitate this in the future, demonstrating that both methods are mutually beneficial and should be promoted.

The computation of gridded-averaged surface current has been proven helpful for the large-scale acquisition of current data. Lilly and Pérez-Brunius (2021a) have used similar analysis and presentation methods for the Gulf of Mexico, averaging spatially and temporally. Averaging over individual deployments spatially is appropriate for the data in this study (Fig. 6). Studies by Coquereau and Foukal (2023) and Qian et al. (2013) show that this is a common method to get an overall view of currents within an area. The strong circulation pattern in the Skagerrak, which is driven by the mixing of saline North Sea water from the Jutland Current (south slope) and fresh water from the Baltic Sea (north slope) (Rodhe, 1996; Christensen et al., 2018), is strongly visible in the gridded representation. It should be noted that this is a seasonal representation. The overall circulation in the surface layer of the Skagerrak is subject to seasonal variations. These are influenced by changes in the Baltic Sea outflow and the Jutland inflow, which in turn is influenced by the discharges of major rivers in the southern part of the North Sea (i.e., Rhine, Meuse, Elbe) (Christensen et al., 2018). This can change the position of the Skagerrak – Baltic Sea front, meaning that the averaging reflects a seasonal snapshot. Nevertheless, the values from the model data in the Skagerrak of up to $0.5$ m s$^{-1}$ show a similar magnitude (Ricker and Stanev, 2020). Although typical values of $0.5 - 0.8$ m s$^{-1}$ increasing occasionally up to $1.0$ m s$^{-1}$ are higher in this present study, these values include a wind-driven flow component and Stokes drift, so a higher value with a small deviation is realistic. The movement of a Lagrangian surface drifter depends on the Eulerian surface current, Stokes drift, and direct wind drag (Röhrs et al., 2012). Studies by Bosi et al. (2021), Stanev et al. (2019) and Röhrs et al. (2012) have shown that during strong wind events, Stokes drift can assume the same magnitude as the Eulerian current velocity or even exceed it. Breivik et al. (2016) demonstrate that Stokes drift exponentially decreases with depth. Röhrs and Christensen (2015) show that for the Barents Sea and the Norwegian Sea, at a mean wind speed of $7.5$ m s$^{-1}$, Stokes drift at the surface is $0.089$ m s$^{-1}$, and at one meter depth is $0.037$ m s$^{-1}$. Our drifter with a size of $0.5$ m would provide an intermediate value. Therefore, for a detailed analysis of residual currents from an Eulerian perspective, a calculation and elimination of Stokes drift will be essential.

In addition, the drifter trajectories in the Skagerrak in Fig. 6b also demonstrate small-scale processes between the inflow of the Jutland Current and the outflow of the Norwegian Coastal Current. This shearing and recirculation indicated by the drifter trajectories are likely induced by geostrophic currents or the convergence of drifters caused by ocean fronts as the power density spectra rule out a pronounced impact of tidal currents and inertial oscillations. The highly temporal resolved drifter data confirms the studies by Lumpkin et al. (2017) and Özgökmen et al. (2012) that drifter data provide the opportunity to

understand the transition processes between the mesoscale ocean circulation and the small-scale fluid dynamics. The study by Lilly and Pérez-Brunius (2021a) complements that large-scale and long-term fluctuations are captured, as well as small-scale and rapid motions. Capturing submesoscale dynamics is challenging with ship-based measurements alone due to the direct

influence of the vessel on the surface layer and fine-scale structures. Remote sensing data often lacks in resolution, and floats and gliders are not able to resolve the temporal scales at the submesoscales due to their measurement principles and, in addition, often exclude the ocean surface boundary layer (McWilliams, 2019). In this context, it is important to capture these dynamics because increasing evidence shows that accounting for the influence of submesoscales on oceanic energetics and tracer fluxes, can improve the fidelity of global ocean and climate models (Taylor and Thompson, 2023). Thus, the Skagerrak dataset is

valuable for the further development of models that cover processes in the submesoscale range. Studies by Graham et al. (2018) and Tonani et al. (2019) show that models with a resolution of 1.5 km already exist for the North Sea, but currently only half of the Skagerrak can be covered by the model. Fig. 6 illustrates that high coverage of high resolution drifter data can fill this gap of knowledge.

Furthermore, this study demonstrates the application of power spectral density to analyze tidal dynamics. The decomposition

of velocity time series into their frequency domain is an established method. Meyerjürgens et al. (2020) used a Fourier transform to represent the frequency domain, which confirms the dominant $M_2$ tidal component for the North Sea, as well as the increased amplitudes for the frequencies of the shallow water components $M_4$ and $M_6$ and the solar semidiurnal $S_2$ component in the southern region of the North Sea. For comparison, a Fourier transform on the drifter dataset is also shown in the supplements (Fig. S1). In this study, power spectral density was chosen because of the additional insights to obtain the

energy of a signal. Similarly, this approach supports the averaging of several time series as a frequency range of length of $2^x$ is generated and all frequency ranges are multiples of each other independent of the signal length.

The analysis of tidal ellipses has also been used for time series of current velocities. Vindenes et al. (2018) used tidal ellipses for an analysis of the northern North Sea using current data from ADCPs. It confirmed that the $M_2$ tide is the most dominant component and that similar magnitudes are obtained for the Skagerrak. The occurrence of the tidal ellipses of $K_1$ and $O_1$ cannot

be confirmed based on the results of power spectral densities, since no peaks emerge from the power spectral densities. It should be noted that Vindenes et al. (2018) provides very small tidal amplitudes for $K_1$ and $O_1$, which may have been masked by the noise in the power spectral densities. Furthermore, the measurements were made at different depths. This present study considers tides at the surface, whereas Vindenes et al. (2018) measured at a depth of 53 m, which may lead to different results. In addition, a small number of time series in the Skagerrak region was used for tidal analysis, because not many trajectories

reached the required minimum trajectory length. Thus, no exact values but trends for the tidal values in the Skagerrak can be provided. However, the analysis method confirms the results by Otto et al. (1990), Ricker and Stanev (2020) and Meyerjürgens et al. (2019) that tidal currents are more pronounced in the shallow southern area of the North Sea (Fig. 7). The magnitudes for the maximum velocities during spring tides can also be confirmed, of up to 1 m s$^{-1}$ in the shallow southern area of the North Sea, up to 0.1 m s$^{-1}$ in the deeper northern area (Dietrich, 1950; Otto et al., 1990), and magnitudes of 0.01 m s$^{-1}$ in the Skagerrak

(Danielssen, 1997; Rodhe, 1987). In addition, the orientation of the southern tidal ellipses shows the distinct east-west motion

in the shallow area of the North Sea, which changes to a north-south motion toward the deeper North Sea, consistent with the description of Otto et al. (1990).

## 6 Summary and Conclusion

The results of the Lagrangian dataset provide valuable insights into current and tidal dynamics in the North Sea region. The calculated values are in agreement with models and in situ measurements, and provide an area-covering measurement method in contrast to Eulerian methods and a higher resolution in contrast to remote sensing and satellite methods. A number of computational methods have been presented that can provide a gridded overview of mean residual circulations and tidal dynamics in the North Sea, which can also be used for other areas. It is important to note that accurate tidal and current analysis can only be generated if measurements are made over long time periods shown in this study for the North Sea. The results clearly distinguish current patterns between the deep part of the Skagerrak and the southern shallow area of the North Sea. The southern shallow water shelf area is strongly influenced by the dynamics of the tides. In contrast, the deep area of the North Sea shows a minor contribution by the tides to the circulation, but strong transport processes by the residual circulations are dominant, which are influenced by high-density gradients.

In the future, these measurements can be complemented by further drifter deployments to provide even higher resolution for current and tidal analysis for accurate values, to detect long-term variations, and to cover a larger spatial domain.

In addition, the sensory developments of the drifter are beneficial. For example, the sampling rate can be adapted to the deployment strategy, depending on the drifter deployment duration or whether mesoscale or submesoscale processes are analyzed. Furthermore, due to the fine sampling rate and by using additional measurement parameters, such as temperature and salinity, the sensory developments could support for example the analyses of ocean front and slick dynamics, as well as weather forecast models.

The dataset should be used for further analysis to provide additional insights for various marine research topics. Global marine litter transport models that previously excluded the North Sea due to the lack of data (Van Sebille et al., 2012; Maximenko et al., 2012) can be extended and areas of particle accumulation can be located. The dataset can be used to determine eddy kinetic energy and other submesoscale processes, and likewise, analysis methods from well-studied regions, such as the Gulf Stream region (Lilly and Pérez-Brunius, 2021b; Oscroft et al., 2020), can be applied to the North Sea region.

## Data Availability

The drifter dataset is available at https://doi.org/10.1594/PANGAEA.963166 (Meyerjürgens et al., 2023b). The dataset contains the interpolated drifter position data, the drifter-derived current velocities, and two vectors ("beached" and "filled") describing when a drifter was beached and when the dataset was filled because no GPS position was sent for more than one hour.

## Funding

This study was carried out within the projects "Macroplastics Pollution in the Southern North Sea-Sources, Pathways and Abatement Strategies" (grant no. ZN3176) and "Sailing Intelligent Micro Drifter Swarms" (grant no. ZN3685) funded by the German Federal State of Lower Saxony, "Carbon Storage in German Coastal Seas - Stability, Vulnerability and Perspectives for Manageability" (grant no. 03F0875C) funded by Federal Ministry of Education and Research and "Biogeochemical processes and Air-sea exchange in the Sea-Surface microlayer" (grant no. 451574234) funded by German Research Foundation.

## Author Contribution

LD performed to the data analysis. LD and JM prepared the draft of the manuscript. JM and THB proposed the scientific idea of the manuscript and performed the data acquisition. All authors contributed to the critical revision of the article and provided important advice for the improvement of the manuscript.

## Conflict of Interest

The authors declare that they have no conflict of interest.

## Acknowledgments

We would like to thank the master and crew onboard the RV Heincke HE503, HE520, HE527, HE583, HE586, and Alkor AL523 for supporting the deployments of the drifters. This allowed the collection of large datasets. We thank Axel Braun, Ulf Harksen, Andreas Sommer, Elisa Janssen, Michael Butter, and Lars Meyer-Hagg for their support in constructing the drifters.

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
