# Peer review of "Figure S1: Mean velocities of the residual circulations for (a) the zonal component $u$ and (b) the meridional component $v$ . The arrows indicate the direction of the mean velocities, the magnitude describes the strength of the velocity. The spring 2018 dataset is included and shows unusual"

_Earth System Science Data, 2023_

## Author Comment (AC1)

*Thank you very much for taking the time to review our manuscript carefully, and we appreciate your constructive and positive feedback. We hereby provide a revised version of our manuscript and provide answers to all comments below. The answers to the comments are in* blue.

Altogether, the paper is written in a clear and concise manner. Also, the English language is used appropriately. Moreover, the surface drifter data are definitely worth to be published, since the provide a complementary view of the ocean surface circulation, when compared with data from mooring stations or from model simulations.

However, at present the paper shows a severe weakness, which should be accounted for, before it is suitable for publication in ESSD.

The benefit of the gridded data is extremely questionable. Hence, its calculation and presentation make no sense, and therefore, it should be removed from the manuscript. There are three major reasons for this criticism:

1)     Mesoscale dynamics occur in a spatio-temporal space, and hence, a temporal average over several months or even years, as done in this study, smooths out most of the mesoscale-structures. Looking at the 15 km resolution gridded data in Figure 8, this problem becomes evident. In fact, only the very coarse cyclonic Skagerrak Gyre circulation is visible. This averaging problem becomes even more obvious, when looking at the finer resolution results. It is apparent, that the 7.5 km, 5 km, and 3 km resolution results do not provide any significant additional information. When looking at the higher resolution results, only the impact of the interpolation scheme can be observed.

Thank you for raising this point. We have reconsidered this point and agree that averaging over the entire time period is only feasible if temporal averaging is also carried out. Seasonal events and mesoscale structures would be excessively smoothed. For this reason, we decided to remove Fig. 2, which showed the residual current for the entire North Sea. The attached figure illustrates that analyses for different time windows, as carried out by Lilly et al. (2021), are not possible because the dataset for the North Sea is limited in this respect.

However, we have retained this approach for the Skagerrak. These data were obtained from one seasonal deployment (Fig. 6, lines 250-263). In principle, averaging the data over a larger area is a common method as seen in Coquereau and Foukal (2023) or Qian et al. (2013). Nevertheless, we have adjusted the figure. For better transparency, data gaps have been removed from the figure and represented as NaN (not a number). In addition, the number of observations has been added to provide better statistical information (lines 264-269). This demonstrates that the data was exclusively averaged, gridded and not interpolated. Thus, a complete overview for the Skagerrak can be provided for this period. The high resolution current velocities in the Skagerrak also present that, with the high resolution position data from drifters, an analysis of submesoscale dynamics is feasible (lines 317-330 and 338-345).

[Figure]

Percentage of Monthly Observations Sampled

2) The argument that compared to model simulations, these gridded data with a 0.125 degrees resolution can provide an improved understanding with respect to the mesoscale dynamics in the North Sea is not acceptable. First, the severe problem mentioned above regarding the temporal averaging of different mesoscale patterns does not occur in model simulations. If required, models do provide results with a temporal resolution even on the subtidal timescale. Moreover nowadays, the spatial resolution of most North Sea models is of the order of 3 km (see e.g. Paetsch et al., 2017), which is nearly one order of magnitude better than the standard resolution of the gridded data, which are presented in this study.

Thank you for the comment. We have revised our results and discussion in this regard. Models and Lagrangian measurements both have their validity, are needed and are essential for future-oriented research. We would like to emphasize that both benefit from each other. We hope this is now formulated more understandably to clarify our statement (lines 308-316).

3)  When converting Lagrangian data to a Eulerian framework you always face the theoretical problem, of how to deal with the Stokes drift, which is inherently included when averaging Lagrangian data over the wind wave scale and/or the tidal scale; both is actually done in the current study. Hence, a discussion about the treatment of the Stokes Drift, when converting drifter data to a Eulerian gird, is definitely necessary. However, at present, this issue has been totally ignored by the authors.

Thank you for raising this point. We have included this point in the discussion because it is undoubtedly an important aspect in the future in order to be able to make detailed statements about residual currents in the North Sea (lines 328-337). A comprehensive analysis of Stokes drift would be beyond the scope of the paper, especially since ESSD is a data journal and the focus should remain on the dataset.

Having in mind these severe problems mentioned above, it is clear that all aspects related to the gridding of the drifter trajectories must be deleted in the manuscript. In contrast, the authors should focus on the real advantages of their very attractive drifter data set. Firstly, the data are an excellent source for model validation. Single drifter trajectories are ideal subjects to be compared with model tracer trajectories in a one-to-one comparison study, using the same starting point and the same time period in retrospective tracer simulations. Secondly, drifter data can nicely be used to derive dispersion properties, as for example performed in Ricker et al. (2022). Since, ideally in hydrodynamical simulations the dispersion rate has to be calibrated for each specific model area, this kind of independent dispersion information can be very helpful for numerous model investigations.

Thank you for pointing this out. We have decided that a detailed validation analysis with models would beyond the scope of the paper due to the fact that this is a data paper. Nevertheless, we also included this in the discussion to highlight the potential of the dataset for model validation (lines 308-316). We have also emphasized the potential of the high resolution dataset. It provides area-wide in situ measurements, which for example provide the opportunity to understand and analyze the transition processes between the mesoscale ocean circulation and submesocale dynamics or the pathways of marine litter (lines 297-305).

---

## Author Comment (AC2)

*Thank you very much for carefully reading our manuscript and for the constructive feedback, which helps us to improve the quality of the manuscript. We addressed all the points raised in the review. We will provide a point-by-point response to each comment below. The responses to the comments are in blue.*

General comments:

The data set presented here is definitely valuable for a detailed insight into the surface currents on different temporal and spatial scales and should be published. However, the manuscript itself and analysis presented therein have a number of weaknesses. Thus, altogether I suggest major revisions of the manuscript.

1. The English language, in some parts, is not satisfying. I recommend to work through the whole text with a native speaker.

Thank you for raising this point. We have carefully revised the English language of the manuscript.

1. The presentation of the used instrument and data set is too scarce. Of course, in Meyerjürgens et al., 2019, a detailed description of the development of the instrument, the measurements and the sampling is given. Yet, Meyerjürgens et al., 2019 is only based on a subset of the data presented here and the reviewed text should stand on its own, as far as the basic information is concerned.

For example:

Determining the position of the drifter and transmitting the position data ashore are key components of the measurements. However, these two components are not correctly described in the text (see lines 129 to 130 and for comparison Meyerjürgens et al., 2019 "Positioning and Telemetry".

Thank you for the comment. We have added a section describing the housing and the components of the drifter in more detail. The used GPS tracker and the satellite transmission are also explained (lines 115-126).

The picture of the instrument in this manuscript (Figure 2) is less helpful than the pictures of the instrument in Meyerjürgens et al., 2019 (Figure 2).

Thanks for pointing this out. We have changed Fig. 2. The drifter is shown as a drawing, as a photo and during a deployment. The main components of the drifter have also been added (lines 112-122).

A reflection on the seasonal coverage of the data set is missing. On top of that, other statistical information such as the number of individual measurement from which the box-averages (Figure 5) were calculated is missing.

The manuscript refers to Lilly and Perez-Brunius, ESSD 2021, which is a good example for how to present a data set of this kind.

Thank you for the comment. For a general overview and statistical information on the dataset, we have added an illustration of the number of observations (Fig. 5, lines 246-249). In the section where the Skagerrak is considered in detail, we have also added the number of observations to ensure better transparency of the current velocity data (Fig.6, line 264/265).

We have decided that we do not want to use a seasonal reflection like Lilly and Perez-Brunius (2021), as our dataset differs in this respect. The dataset for the Gulf of Mexico includes measurements from more than 28 years, so it is much larger and evenly distributed. This point has not yet been reached in the North Sea (see attached figure). Our dataset, on the other hand, provides high resolution coverage for some areas in the North Sea, or for seasonal events, and can be used for model validations and offers the possibility to analyze submesoscale dynamics (lines 308-316 and 338-345).

[Figure]

For this reason, we have decided to remove the analysis of residual currents across the entire North Sea from the paper. We agree that for an overall view, in addition to spatial averaging, the temporal view and seasonal fluctuations must also be considered. However, the transfer to an Eulerian approach should remain part of the paper, but only for seasonal deployments such as the one in the Skagerrak (Fig. 6). The approach for individual deployments is common, see for example Coquereau and Foukal (2023) or Qian et al. (2013). The results and the discussion were largely revised (lines 250-269 and 317-345).

1. When dealing with different scales of motion, I expect at first some reflections on the time and length scales of the different processes in the research area and considerations if and how these processes can be observed with the data set at hand.

See for example Cushman-Roisin and Beckers, Introduction to geophysical fluid dynamics, Elsevier 2011, or search for "baroclinic Rossby radius".

It is definitely not possible to sort out different scales by only decreasing the grid-cell length!

Thanks for the comment. As mentioned above, we have largely revised our approach to clarify our statement and intention. We have supplemented the number of observations (lines 246-249), removed the figure with the mean residual currents for the entire North Sea and significantly revised the results and discussion (lines 250-269 and lines 297-345).

We have continued to present the currents in the Skagerrak, as these originate from a seasonal deployment. We did not want to eliminate individual scales with the different resolutions. We chose different resolutions to emphasize that the dataset offers a high area-wide resolution and that the submesocale processes can better be represented with a high resolution. The high resolution dataset offers the possibility to analyze submesoscale dynamics.

I will not go into detail for this part of the manuscript but also go along with the comments from reviewer #1.

1. Figure 5 a/b: Although a and b are meant to show the zonal/meridional component of the velocities, the arrows are similar in both pictures. I recommend having only one picture and show in colors the magnitude of the absolute value of the velocity and the direction; red – eastward, blue – westward.

Thanks for pointing this out. As we have decided against averaging and gridding (see above) the figure with the currents for the entire North Sea have been removed.

Individual comments:

English language:

Line 45: I suggest "..consisted of mooring arrays with…" instead.

We have revised it as suggested (line 45).

Line 65/66: I don't understand the logical implication of ".. resulting in.." in this sentence. Please rewrite.

We have revised the paragraph regarding your comment (lines 65-69).

Line 68: I don't understand the logical implication of "..due to.." in this sentence. Please rewrite.

We have revised the paragraph regarding your comment (lines 65-69).

Line 92: I don't understand the logical implication of "In addition" in this sentence. Please rewrite.

We have rephrased the sentence to clarify our statement (line 90/91).

Line 92: "These are partly due to nonlinear tidal interactions." I don't understand this statement. Please explain and rewrite.

We have revised the paragraph to clarify our statement. A nonlinearity occurs due to nonlinear advection terms in shallow coastal regions and bottom friction (lines 90-95).

Line 122: I suggest "..and measured for up to several month".

We have removed this part (line 134).

Line 180: "…a grid is created that is composed of several cells." I have no idea of what else a grid can consist of. This part of the sentence give the same information two times. Please rewrite or rather shorten.

As we have decided against averaging and gridding (see above), with the exception of seasonal deployments (e.g. Skagerrak), this paragraph and sentence in the methods section have generally been removed.

Content:

Line 31: The North Sea is also influenced by river-runoff from Forth, Humber, Thames, Seine, Meuse, Glomma. See for example "North Sea Region Climate Change Assessment", Editors Markus Quante and Franciscus Colijn, Springer 2016. DOI 10.1007/978-3-319-39745-0.

We have revised it as suggested and added the citation (line 31/32).

Line 36/37: The residual currents are driven by density gradients and wind.

We have revised it as suggested (line 37).

Line 51-53: This is basically correct, but whether or not the data set is able to resolve small scales of motion depends on the time increment of the measurements. And this also holds for Eulerian measurements. This part is not really describing the difference between Lagrangian and Eulerian measurements. Please rewrite.

Thanks for pointing this out. We have revised this paragraph to clarify our statement (line 51/52).

Line 75. Please write southwest instead of south.

We have revised it as suggested (line 72).

Line 81-83: Please add a reference.

We have added the references of Otto et al. (1990) and Vindenes et al. (2018) (lines 79-81).

Line 87: I suggest giving all numbers in m/s and not changing the units within the paper.

We have revised it as suggested.

Line 93-94: Interaction needs different components: please specify.

We have revised the paragraph to clarify our statement. A nonlinearity occurs due to nonlinear advection terms in shallow coastal regions and bottom friction (lines 90-95).

Line 103-104: "… the residual circulations are assumed to be smaller compared to tidal currents and wind-induced currents". Please rewrite this sentence, because wind-induced currents are part of the residual currents.

We have rephrased the sentence to clarify our statement (line 102).

Line 114: Please specify why this drifters can be deployed in shallow water and others cannot.

We have revised the text according to your comment (lines 123-126).

Table 1: Is the number of trajectories similar to the number of instruments?

Yes it is. To clarify our statement, we added in the table header: "number of drifter trajectories" and added in line 140/141, that we measured the 85 trajectories with 85 surface drifters.

Line 134: I don't understand the terminus "typical circulation trend" – do you mean "mean residual circulation"? Please explain.

We have rephrased the sentence to clarify our statement (line 146).

Line 135: It should be: "moved westward".

You are right, please excuse the mistake. We changed it (line 147).

Line 173: I have never heard about "the complex-valued velocity". The vector of the surface velocity consists of a zonal and a meridional component. But why is it complex? Please explain.

We have removed "complex-valued". Mathematically, it is used to facilitate the calculation of the velocity data. However, you are right that from the physical perspective presented here, a "complex-valued velocity" does not exist.

Line 188/89: How many data gaps (percentage) in the grid have been filled by this method?

To improve the transparency, we have decided to remove the data gaps and present them as NaN (not a number) (Fig. 6, line 255/256). It can be seen that the drifter measurements covered most of the Skagerrak.

Line 211 and 220: Please explain why you need a time series of "at about 14.2 days" and "at least 10.4 days"?

Thank you for pointing this out. We have added comparable studies and revised the sections (lines 208-217 and 225-228). In principle, time series are required where distinct peaks can be seen in the analysis in order to obtain statistically significant results. This is reached for our dataset after 10.4 days (3000 measurements).

For the power spectral density we took slightly more measurements (14.2 days = 4096 measurements), as a power of 2 is required for the calculation ($2^{12}$ = 4096).

Line 243/244: I think "which results in" is the wrong logical implication.

Additionally, you can have "long residence times" in a grid cell, if velocities are small. Thus, you have to explain this statement or show it is proven.

We have removed this section (see above).

Line 248-252: The naming "eastern inflow of the North Atlantic" is not correct, especially if you are dealing with surface currents.

Thank you for the comment. We changed the inflow to the Jutland current and added the reference Christensen et al. (2018) (lines 240-269 and 321-326).

I miss some information about variability of the features "inflow of the North Atlantic", "Norwegian Coastal Current" and "outflow from the Baltic Sea". They are highly variable and thus averaging about all data in a grid cell, independent of the time resolution/coverage can produce artefacts (more information about this topic/problem can be found at Lilly and Perez-Brunius, ESSD 2021 and Lilly and Perez-Brunius, Nonlin. Processes Geophys, 2021 from your references).

We have added a small section to the discussion (lines 321-326).

I don't see any "small eddies" in the map (Figure 5).

We have revised the text according to your comment (line 260-269).

Line 260/261: "..which means that the north-south motion increases". This is a change in direction of only 6%. I suggest to write "slightly increases".

We have revised it as suggested (line 279).

Line 260: see Line 87.

We have revised it as suggested.

Line 270/271: Please explain or rewrite: "Other irregularities are difficult to distinguish from the rest of the noise". What do you mean with "the rest of the noise"?

We have revised the sentence to clarify our statement (line 289/290).

5 Discussion, first paragraph: It is difficult to distinguish between general statements, taken from the literature, and results from the analyses of the presented data set. Please clarify. From this study we do not "gain an understanding of the transition processes between the mesoscale ocean circulation and" smaller scales.

We have revised the paragraph in order to better distinguish our own results, general statements and the literature (lines 297-316). We rephrased the sentence to clarify our statement (lines 300-303).

---

## Author Response (AR2)

*Thank you very much for carefully reading our manuscript, we are pleased that we have already been able to improve the manuscript. Thank you for your further constructive feedback. This helps us again to increase the quality of the manuscript. We have addressed all the points raised in the review. The responses to the comments are in* blue*. The lines refer to the revised, resubmitted manuscript (without tracking changes).*

The paper has been strongly improved due consideration of the comments of both reviewers. There is just one general criticism left. It concerns Fig. 6. I do not see that the 5 km resolution shows any additional mesoscale features compared to the 7.5 km resolution as stated in lines 266-299. The authors refer to a small-scale shearing, which I am unable to see. Speaking of additional mesoscale features, I would expect to see additional smaller-scale eddies etc., which are obviously are not visible.

To my understanding these mesoscale features are missing due to two possible reasons

1)  In the period, which was investigated, the Skagerrak area is too strongly dominated by the basin-wide Skagerrak Gyre, and therefore no additional smaller scale features are observable.

2)  An averaging over an entire season smooths out all small-scale features, which would be visible on a snapshot in the Eulerian framework. In the reply to my comment no. 1 the authors defended their proceeding, by mentioning that an averaging over a larger area is a common procedure. I can fully accept this point. However, my suspicion is, that the temporal averaging causes the major problem in this case, since in general, mesoscale features are time-dependent, non-stationary phenomena.

In both circumstances, i.e., the Skagerrak area does not show significant mesoscale features or the temporal averaging destroys all mesoscale features, the presentation of Fig. 6 is not meaningful, and hence, should be omitted.

Besides this issue mentioned above, I have no further concerns. And I would fully recommend the publication of this manuscript in ESSD.

Thank you very much for your comments.

We have reconsidered this aspect and agree that the temporal averaging largely smoothes out the submesoscale processes. For this reason, we have adjusted the figure. First, we removed the comparison of the different resolution and added the trajectories to illustrate finer processes (lines 248-266). We think it is important that the Skagerrak is still presented, as the uniqueness of the dataset should be emphasized. The currents in the Skagerrak area differ significantly from the rest of the North Sea, as tidal effects hardly influence them but are strongly influenced by highly dynamic density-driven current processes (lines 315-331). In addition, the high resolution data enable the further development of models that cover submesoscale processes and can be used for further analyses of submesoscale processes (lines 332-348).

We have also adjusted our title accordingly because the focus has changed. The high resolution current data are now more emphasized and not the submesoscale processes.